# Innovative tracking, active search and follow-up strategies for new leprosy cases in the female prison population

**Claudia Maria Lincoln Silva**[1,2☯], **Fred Bernardes Filho**[1,2☯], **Glauber Voltan**[1,2☯], **Jaci Maria Santana**[1,2☯], **Marcel Nani Leite**[1,2☯], **Filipe Rocha Lima**[1,2☯], **Luisiane de Avila Santana**[1,2☯], **Natália Aparecida de Paula**[1,2☯], **Patricia Toscano Barreto Nogueira Onofre**[3], **Wilson Marques-Junior**[3], **Vanessa Aparecida Tomaz**[4], **Carmem Sílvia Vilela Pinese**[4], **Marco Andrey Cipriani Frade**[1,2☯]*

1 Dermatology Division, Department of Medical Clinics, Ribeirão Preto Medical School, University of São Paulo, Ribeirão Preto, São Paulo, Brazil, 2 Center of National Reference in Sanitary Dermatology focusing on Leprosy of Ribeirão Preto Clinical Hospital, Ribeirão Preto, São Paulo, Brazil, 3 Department of Neuroscience and Behavioral Sciences, Ribeirão Preto Medical School, University of São Paulo, Ribeirão Preto, São Paulo, Brazil, 4 Ribeirão Preto Female Penitentiary, Penitentiary Administration Secretariat, Ribeirão Preto, São Paulo, Brazil

☯ These authors contributed equally to this work.
* mandrey@fmrp.usp.br

**Data Availability Statement:** All relevant data are within the manuscript.

## Abstract

### Background

Regarding the leprosy transmission through the upper airways, overcrowded locations such as prisons can become a risk to get sick. Like the leprosy hidden endemic demonstrated in male prison population, being interesting to assess the leprosy scene also among confined women.

### Methods

A prospective descriptive study conducted at Female Penitentiary, Brazil. Leprosy Suspicion Questionnaire (LSQ) were applied to the participants, and submitted to specialized dermatoneurological exam, peripheral nerve ultrasonography, and anti-PGL-I serology.

### Findings

404 female inmates were evaluated, 14 new cases were diagnosed (LG-leprosy group), a new case detection rate (NCDR) of 3.4%, 13 multibacillary, while another 390 constituted the Non-Leprosy group (NLG). Leprosy cases were followed up during multidrug therapy with clinical improvement. The confinement time median was 31 months in LG, similar to NLG, less than the time of leprosy incubation. Regarding LSQ, the neurological symptoms reached the highest $x^2$ values as Q1–numbness (5.6), Q3–anesthetizes areas in the skin (7.5), Q5–Stinging sensation (5.8), and Q7–pain in the nerves (34.7), while Q4-spots on the skin was 4.94. When more than one question were marked in the LSQ means a 12.8-fold higher to have the disease than a subject who marked only one or none. The high 34% rate

**Funding:** This study was supported by the following grants: MACF - Center of National Reference in Sanitary Dermatology focusing on Leprosy of Ribeirão Preto Clinical Hospital, Ribeirão Preto, São Paulo, Brazil; MACF - Brazilian Health Ministry (MS/FAEPAFMRP-USP: 749145/2010 and 767202/2011); MACF -Fiocruz Ribeirão Preto - TED 163/2019 - Process: N˚ 25380.102201/2019-62/ Projeto Fiotec: PRES-009-FIO-20. The funders had no role in study design, data collection and analysis, decision to publish, or preparation of the manuscript.

**Competing interests:** The authors have declared that no competing interests exist.

of anti-PGL-I seropositivity in the penitentiary, higher levels in LG than NLG. Three additional leprosy cases each were diagnosed on the second (n = 66) and third (n = 14) reevaluations 18 and 36 months after the initial one. Semmes-Weinstein monofilaments demonstrated lower limbs (32.2%) more affected than the upper limbs (25%) with improvement during the follow-up.

## Interpretation

The NCDR in this population showed an hidden endemic of leprosy as well as the efficacy of a search action on the part of a specialized team with the aid of the LSQ and anti-PGL-I serology as an auxiliary tracking tools.

### Author summary

The study was conducted in a Female Penitentiary-Brazil. The Leprosy Suspicion Questionnaire (LSQ) was administered to all the inmates who also underwent, specialized dermato-neurological exam, and anti-PGL-I serology. We evaluated 404 female inmates, 14 new cases were diagnosed (LG-leprosy group), and 390 constituted the Non-Leprosy group (NLG), a new case detection rate of 3.4%. The confinement time was shorter than the time of leprosy incubation. Leprosy cases were followed up during multidrug therapy and it showed clinical improvement. Regarding the LSQ, the most important neurological symptoms were Q1–numbness, Q3–anesthetized areas, Q5–Stinging sensation, and Q7–pain in the nerves, while Q4-spots were fewer. When more than one question was marked, it meant a 12.8 higher chance to get leprosy. The high anti-PGL-I seropositivity among females was higher in the LG than in the NLG. Afterwards, we reevaluated the inmates and 6 additional leprosy cases were diagnosed in two visits. Semmes-Weinstein monofilaments demonstrated that the lower limbs (32.2%) were more affected than the upper limbs (25%) with improvement during the follow-up. Thus, our study in this population showed hidden leprosy as in the male prison. It also showed how efficient the search by a specialized staff with the aid of LSQ, as an auxiliary tracking tool, is.

## Introduction

Leprosy is a chronic disease characterized by neurological and dermatological manifestations that lead to disability [1]. It is caused by *Mycobacterium leprae*, a bacillus of high infectivity and low pathogenicity [2] and it is mainly associated with social inequality, affecting the most underprivileged areas in the world [3]. Transmission occurs through the upper airways of the patients with a high bacillary load, depending essentially on the relationship with the host and the degree of endemicity of the environment [4, 5]. Unhealthy conditions can favor the endemicity of leprosy [6].

The world prison population is estimated to be approximately 12 million [7]. Currently, Brazil occupies the third place in the ranking of the 10 countries with the highest prison population (752,277) and it has the fourth largest female prison population in the world (37,828), which means that women represent 6.4% of the Brazilian prison population [7, 8].

The overcrowding and precarious conditions of the prison cells make this environment favorable to the spreading of the disease [9]. Moreover, poor feeding, lack of hygiene,

sedentarism and the use of drugs, among other conditions, contribute to the occurrence of the disease in this population [10]. Recent publication about leprosy in a male prison population demonstrated a hidden prevalence of leprosy. The individuals diagnosed were likely to have been infected while living in their former communities and not because of exposure in the prison [11], though.

Within this context and considering the reality seen among incarcerated men, it is also interesting to know the reality of leprosy among women who have been in long periods of imprisonment and the factors associated to its diagnosis and its follow-up.

## Methods

### Ethics statement

This study was approved by the Research Ethics Committee at the Clinics Hospital of Ribeirão Preto Medical School, University of São Paulo (protocol number 2,165,032). Written informed consent was obtained from every participant. All procedures involving human beings comply with the ethical standards of the Helsinki Declaration (1975/2008).

### Type of study

This study was characterized as a prospective clinical and epidemiological study.

### Setting and study population

This study was conducted from September 2017 to November 2020, involving 404 women initially corresponding to the total population of prisoners in a closed regime at the Ribeirão Preto Female Penitentiary.

### Application of the Leprosy Suspicion Questionnaire (LSQ)

The prisoners received a suspicion questionnaire for leprosy containing 14 simple questions and describing significant signs and symptoms of the disease according to Bernardes Filho et al [11]. After receiving guidance and responding to it, the prisoners returned the filled-out questionnaire and, afterwards, they underwent a clinical-dermatological assessment. Questionnaires about habits and customs, mainly regarding social and environmental aspects, were also administered at the time of assessment.

### Diagnostic criteria for leprosy

The subjects underwent a standardized clinical dermato-neurological test according to the Brazilian Ministry of Health guidelines. Leprosy diagnosis was made upon detection of at least one of the following signs/symptoms: a) lesion(s) and/or area(s) of the skin with changes in thermal and/or painful and/or tactile sensitivity; b) thickening of the peripheral nerve(s), associated with sensory and/or motor and/or autonomic changes; and/or c) presence of *M. leprae* confirmed by intradermal smear microscopy or skin biopsy [12]. After certification by at least two experts, two groups were established: individuals diagnosed with leprosy (Leprosy Group, LG) and another one with the other individuals (Non-Leprosy Group, NLG).

### Peripheral nerve ultrasonography

As published before by Frade et al. (2013) [13], we analyzed the cross-sectional areas (CSA) in median nerves (carpal tunnel and distal forearm), ulnar nerves (cubital tunnel and distal arm), common fibular nerves (head of fibula and distal thigh) and tibial nerves (posterior to the

ankles). The nerve asymmetry was calculated by the difference between the biggest CSA and the smallest one in the same nerve point. Nerve focality was calculated by the difference between two points (proximal and distal CSA) in the same nerve. Qualitative morphological alterations were defined by loss of fascicular nerve pattern, heterogeneous fascicular distention, signs of perineural fibrosis.

### Assessment of anti-PGL-I titer by ELISA

Indirect ELISA was used to measure the anti-PGL-I IgM titer of every the serum sample using the protocol previously reported [14]. The sample index was calculated by dividing their optical density (O.D.) by the cut-off; indexes above 1.0 were considered positive.

### DNA extraction and RLEP amplification

Total DNA extraction of earlobes and at least one elbow and/or lesion SSS sample using the QIAamp DNA Mini Kit (Qiagen, Germantown, MD, cat: 51306) was performed according to the manufacturers protocol. DNA was used to performed PCR-RLEP according to Azevedo et al (2017) [15].

### Follow-up of patients

New leprosy cases diagnosed via active detection were followed-up monthly by the research team focused to evaluate the improvement/worsening of the neurological symptoms, and cutaneous signs, and side effects of multidrug therapy (MDT). Hands/feet tactile sensitivity tests (Semmes Weinstein monofilaments) were carried out on the diagnosis, in the 6th month and in the end of treatment.

### Statistical analysis

All data were analyzed with GraphPad Prism v.7.0 software (GraphPad Inc., La Jolla, CA, USA). Statistical differences were analyzed by the Mann-Whitney test for the comparison of anti-PGL-I immunoglobulin levels. The chi-square test was used to assess associations among categorical variables. The z-score test was used to calculate the difference between two-population proportions. The level of statistical significance was set at $p < 0.05$ in all analyses.

## Results

A total of 404 women were initially evaluated. At first, all of them received the Leprosy Suspicion Questionnaire (LSQ) and all of them filled it out and returned it. A total of 250 (61.9%) questionnaires were considered positive (LSQ+) for one or more signs/symptoms, and distributed according to both groups as shown in Table 1.

After undergoing dermato-neurological clinical examination by the team, fourteen new leprosy cases were diagnosed among the 404 women initially evaluated, resulting in a 3.5% new case detection rate (NCDR) at the Ribeirão Preto Female Penitentiary.

The subjects were divided into two groups: the Leprosy Group (LG, n = 14, average age: 33 years old), and Non-Leprosy Group (NLG, n = 390, average age: 32.5 years old), with no difference ($p > 0.05$). Regarding the time of confinement, the two groups were also similar, average of 31 months in LG and 29 months in NLG ($p > 0.05$).

All LG women reported positivity for one or more of signs/symptoms of the disease when answering the LSQ, with a mean of 4 positive responses (range: 1–9), while in the NLG, 236 (60.51%) women replied to the LSQ+, with a mean of 3 positive responses (range: 1–10), as described in Table 1.

**Table 1. Number of individuals ranked according to total signs and symptoms of leprosy marked on the LSQ in order of frequency (n = 250).**

| Q | Leprosy Suspicion Questionnaire (LSQ) | Total (n) | % | NLG (n) | % | LG (n) | % | $X^2$ | Z | p |
|---|---|---|---|---|---|---|---|---|---|---|
| | Number of LSQ distributed | **404** | | - | - | - | - | | | |
| | Number of LSQ returned | **404** | 100 | **390** | 96.5 | **14** | 3.5 | | | |
| | Number of LSQ respondees evaluated | **404** | 100 | **390** | 96.5 | **14** | 3.5 | | | |
| | Number of LSQ with some marking (LSQ+) | 250 | 61.9 | 236 | 94.4 | 14 | 5.6 | | | |
| | **Symptoms and Signs (LSQ+)** | | | | | | | | | |
| 1 | Do you feel numbness in your hands and/or feet? | 133 | 53.2 | 124 | 52.5 | 9 | 64.3 | 5.57 | 2.54 | 0.011 |
| 2 | Tingling (pricking)? | 168 | 67.2 | 160 | 67.8 | 8 | 57.1 | 1.73 | 1.20 | 0.230 |
| 3 | Anesthetized areas in the skin? | 43 | 17.2 | 38 | 16.1 | 5 | 35.7 | 7.50 | 3.10 | 0.002 |
| 4 | Spots on the skin? | 72 | 28.8 | 66 | 28.0 | 6 | 42.9 | 4.94 | 2.49 | 0.012 |
| 5 | Stinging sensation? | 87 | 34.8 | 80 | 33.9 | 7 | 50.0 | 5.77 | 2.63 | 0.008 |
| 6 | Nodules on the skin? | 80 | 32.0 | 79 | 33.5 | 1 | 7.1 | 0.66 | 1.21 | 0.226 |
| 7 | Pain in the nerves? | 48 | 19.2 | 39 | 16.5 | 9 | 64.3 | 34.53 | 6.17 | <0.001 |
| 8 | Swelling of hands and feet? | 89 | 35.6 | 84 | 35.6 | 5 | 35.7 | 1,017 | 1.26 | 0.210 |
| 9 | Swelling of face? | 25 | 10.0 | 24 | 10.2 | 1 | 7.1 | 0.149 | 0.15 | 0.880 |
| 10 | Weakness in hands? | 55 | 22.0 | 52 | 22.0 | 3 | 21.4 | 0.279 | 0.87 | 0.380 |
| 11 | Hard to button shirt? Wear glasses? Write? Hold pans? | 29 | 11.6 | 28 | 11.9 | 1 | 7.1 | 0.254 | -.005 | 0.992 |
| 12 | Weakness in feet? Difficulty wearing sandals. slippers? | 12 | 4.8 | 11 | 4.7 | 1 | 7.1 | 0.023 | 0.94 | 0.347 |
| 13 | Loss of eyelashes? | 7 | 2.8 | 7 | 3.0 | 0 | 0 | - | -.51 | 0.610 |
| 14 | Loss of eyebrows? | 15 | 6.0 | 14 | 5.9 | 1 | 7.1 | 0,0 | 0.69 | 0.490 |
| | Total number of answers | | | | | | | | | |
| | Mean answers/individual | | | 3 | | 4 | | | | |
| | Min | | | 1 | | 1 | | | | |
| | Max | | | 10 | | 9 | | | | |

Q: question number; NLG: non-leprosy group; LG: leprosy group; LSQ+: Number of LSQ with some mark.

The chi-square test with Yates correction was applied regarding each question individually and the chi-square values of neurological symptoms were: 5.57 for Q1-numbness, 7.5 for Q3-anesthetized areas in the skin, 5.77 for Q5-Stinging sensation and 34.5 for Q7-pain in the nerves, while the chi-square value of the dermatological sign Q4-spots on the skin was 4.94.

Since 100% of the patients showed LSQ+, we opted to separate the number of individuals with more than one response to the LSQ (LSQ>1+) from the number of individuals with one or no response (LSQ≤1+), dividing them into Leprosy and Non-Leprosy groups, as shown in Table 2.

**Table 2. Distribution of individuals between groups according to the number of responses marked given to the LSQ greater than or less than 1 (Table 2 x 2).**

| | LG | LNG | TOTAL |
|---|---|---|---|
| LSQ >1+ | 13 (7.03) [5.06] | 190 (195.97) [0.18] | 203 |
| LSQ ≤1+ | 1 (6.97) [5.11] | 200 (194.03) [0.18] | 201 |
| TOTAL | 14 | 390 | 404 |

LG: leprosy group; NLG: non-leprosy group; LSQ: Leprosy Suspicion Questionnaire; (the expected cell totals) and [the chi-square statistic for each cell].

From this table, a chi-square test of independence with Yates's correction was performed to examine the relation between those with LSQ+ in a group of more than 1 question and the possibility of being diagnosed with leprosy compared to the ones into a group with 1 or less questions marked. The relation between these variables was significant, $x^2$ (1, n = 404) = 8.9, p = 0.0029. The incidence rate in the population was 3.47%, among those in the LSQ>1+ it was 6.40%, while among individuals with LSQ$\leq$1+, it was 0.5%. The relative risk found was 12.87 and odds ratio was 13.7, i.e., an individual with more than one question marked in the LSQ had a 12.8 higher chance of having the disease than an individual who marked only one or no question.

Questions 1, 2, 5, and 7 were those more marked in LG, while questions 1, 2, 4, 5, 7 and 8 were those more marked in NLG.

Four of the six questions more frequently marked referred to the neurological symptoms of leprosy in both in the LG and NLG (p>0.05). Highlighting the importance of the neurological symptoms, their frequency was maintained when we crossed the questions which had more responses, as shown in Table 3.

On the other hand, considering only the associated symptoms which were presented at least in half of leprosy cases, we observed that only associating the neurological symptoms (Q1, Q2, Q5 and Q7), we have reached the highest risks by odds-ratio when Q5-stinging sensation was associated with Q1-numbeness (10.14), Q7-pain in the nerves (8.07), and with Q2-tingling/pricking (7.48) further supporting the importance of neurological symptoms compared to the skin signs for the diagnosis of leprosy.

The specific clinical aspects of leprosy patients i.e. dysesthesia, dysautonomia, involvement of peripheral nerves, hands and feet tactile sensitivity test, classification and presence of a vaccine scar are listed, according to the anti-PGL-I results in Table 4.

Only the hands and feet tactile sensitivities seems to have relation with the anti-PGL-I positivity highlighting the Semmes Weinstein monofilaments (SMW) used to detect these neuropathic signs for the diagnosis of leprosy. Curiously, only 7 (35%) of LG patients presented

**Table 3. Distribution of crossing frequencies between the most marked questions of LSQ involving Q1, Q2, Q5, and Q7 neurological symptoms and respective chi-square, risk relative, odds ratio and p-values (n = 404).**

| Q + Q | LG (n = 14) | | NLG (n = 236) | | $X^2$ | RR | Odds ratio | *p-value* |
|-------|------|------|------|------|-------|-----|------------|-----------|
| | N | % | N | % | | | | |
| Q1 + Q2 | 8 | 3.2 | 97 | 38.8 | 7.32 | 3.8 | 4.03 | 0.007 |
| Q2 + Q7 | 7 | 2.8 | 83 | 33.2 | 6.44 | 3.5 | 3.70 | 0.011 |
| Q1 + Q7 | 7 | 2.8 | 62 | 24.8 | 11.10 | 4.9 | 5.29 | <0.001 |
| Q1 + Q5 | 7 | 2.8 | 35 | 14 | 24.42 | 8.6 | 10.14 | <0.001 |
| Q2 + Q5 | 7 | 2.8 | 46 | 18.4 | 17.31 | 6.6 | 7.48 | <0.001 |
| Q5 + Q7 | 7 | 2.8 | 43 | 17.2 | 18.93 | 7.1 | 8.07 | <0.001 |
| Q1 + Q4 | 4 | 1.6 | 18 | 7.2 | 15.06 | 6.9 | 8.27 | <0.001 |
| Q1 + Q8 | 4 | 1.6 | 42 | 16.8 | 31.54 | 3.1 | 3.31 | <0.001 |
| Q2 + Q8 | 3 | 1.2 | 53 | 21.2 | 0.70 | 1.7 | 1.73 | 0.4000 |
| Q4 + Q5 | 3 | 1.2 | 28 | 11.2 | 3.87 | 3.3 | 3.53 | 0.0490 |
| Q2 + Q4 | 2 | 0.8 | 44 | 17.6 | 0.73 | 1.3 | 1.31 | 0.1208 |
| Q4 + Q8 | 2 | 0.8 | 24 | 9.6 | 1.48 | 2.4 | 2.54 | 0.2231 |
| Q5 + Q8 | 2 | 0.8 | 27 | 10.8 | 1.10 | 2.2 | 2.24 | 0.2940 |
| Q7 + Q8 | 2 | 0.8 | 50 | 20 | 0.03 | 1.1 | 1.13 | 0.8722 |

Q: question number; n: number of leprosy patient; RR: relative risk; *p*: significance value.

**Table 4. Clinical characterization of Ribeirão Preto Female Penitentiary patients regarding the percentage of positivity to the clinical criteria used for the diagnosis of leprosy (n = 14) distributed according to the anti-PGL-I results.**

| Clinical criteria | Yes (n = 14) | % | anti-PGL-I + (n = 10) | % | anti-PGL-I— (n = 4) | % | z | p |
|---|---|---|---|---|---|---|---|---|
| Dysesthesia hypochromic macular skin lesions | | | | | | | | |
| Thermal + tactile + pain sensitivities | 13 | 92.9 | 10 | 76.9 | 3 | 23.1 | 1.64 | 0.10 |
| Tactile + pain sensitivities | 1 | 7.1 | 0 | 0 | 1 | 100 | | |
| Localized irregular patches of circumscribed hair loss | 6 | 42.9 | 5 | 83.3 | 1 | 16.7 | 0.85 | 0.39 |
| Endogenous histamine test | | | | | | | | |
| Incomplete | 13 | 92.9 | 9 | 69.2 | 4 | 30.8 | -0.66 | 0.51 |
| Not performed | 1 | 7.1 | 1 | 10 | 0 | 0 | | |
| Altered nerves on palpation (enlargement and/or pain and/or electric shock-like pain) | 11 | 78.6 | 8 | 72.7 | 3 | 27.3 | 0.21 | 0.83 |
| Hands tactile sensitivity | | | | | | | | |
| Normal (green monofilament 0.05gram-force) | 8 | 57.1 | 4 | 50 | 4 | 50 | | |
| Abnormal | 6 | 42.9 | 6 | 100 | 0 | 0 | 2.05 | 0.04 |
| Foot tactile sensitivity | | | | | | | | |
| Normal (green and blue monofilaments 0.05/0.2 gram-force) | 4 | 28.6 | 0 | 0 | 4 | 100 | | |
| Abnormal | 10 | 71.4 | 10 | 100 | 0 | 0 | 3.74 | 0.0002 |
| Leprosy classification | | | | | | | | |
| Indeterminate | 1 | 7.1 | 1 | 100 | 0 | 0 | | |
| Borderline | 12 | 85.7 | 8 | 66.7 | 4 | 33.3 | | |
| PNL | 1 | 7.1 | 1 | 100 | 0 | 0 | | |
| WHO operational criteria | | | | | | | | |
| Paucibacillary | 1 | 7.1 | 1 | 100 | 0 | 0 | | |
| Multibacillary | 13 | 92.9 | 9 | 69.2 | 4 | 30.8 | | |
| WHO impairment grading | | | | | | | | |
| Grade 0 | 6 | 42.9 | 3 | 50 | 3 | 50 | | |
| Grade 1 | 8 | 57.1 | 7 | 87.5 | 1 | 12.5 | 1.54 | 0.12 |
| Grade 2 | 0 | 0 | 0 | 0 | 0 | 0 | | |
| BCG scar | | | | | | | | |
| 0 | 2 | 14.3 | 2 | 100 | 0 | 0 | | |
| 1 | 12 | 85.7 | 8 | 66.7 | 4 | 33.3 | | |

BCG Bacillus Calmette–Guérin; PNL pure neural leprosy; WHO World Health Organization.

some alterations (does not fell 0.05g-force) in the hands detected by SMW, while 14 (70%) of them presented sensitivity alterations (does not fell 0.2g-force) in the feet (p = 0.003).

For information and recording purposes, the evaluations were photographed and the responses to sensitivity questions were marked with a pen using the following symbols: minus (-) reduced sensitivity; zero (0) absence of sensitivity, and plus (+) for normal sensitivity [16], as illustrated in Fig 1.

Anti-PGL-I ELISA index was positive in both groups, 11 (78.6%) in LG and 151 (38.7%) in NLG (Table 5).

The serum level of anti-PGL-I IgM antibodies was significantly higher (p <0.001) among leprosy patients (LG) than individuals from the NLG.

All patients were monitored monthly during the administration of all doses of the MDT scheme, with clinical dermato-neurological and esthesiometry evaluations. During the fifth month of MDT, one patient presented changes in skin and mucosa pigmentation, with

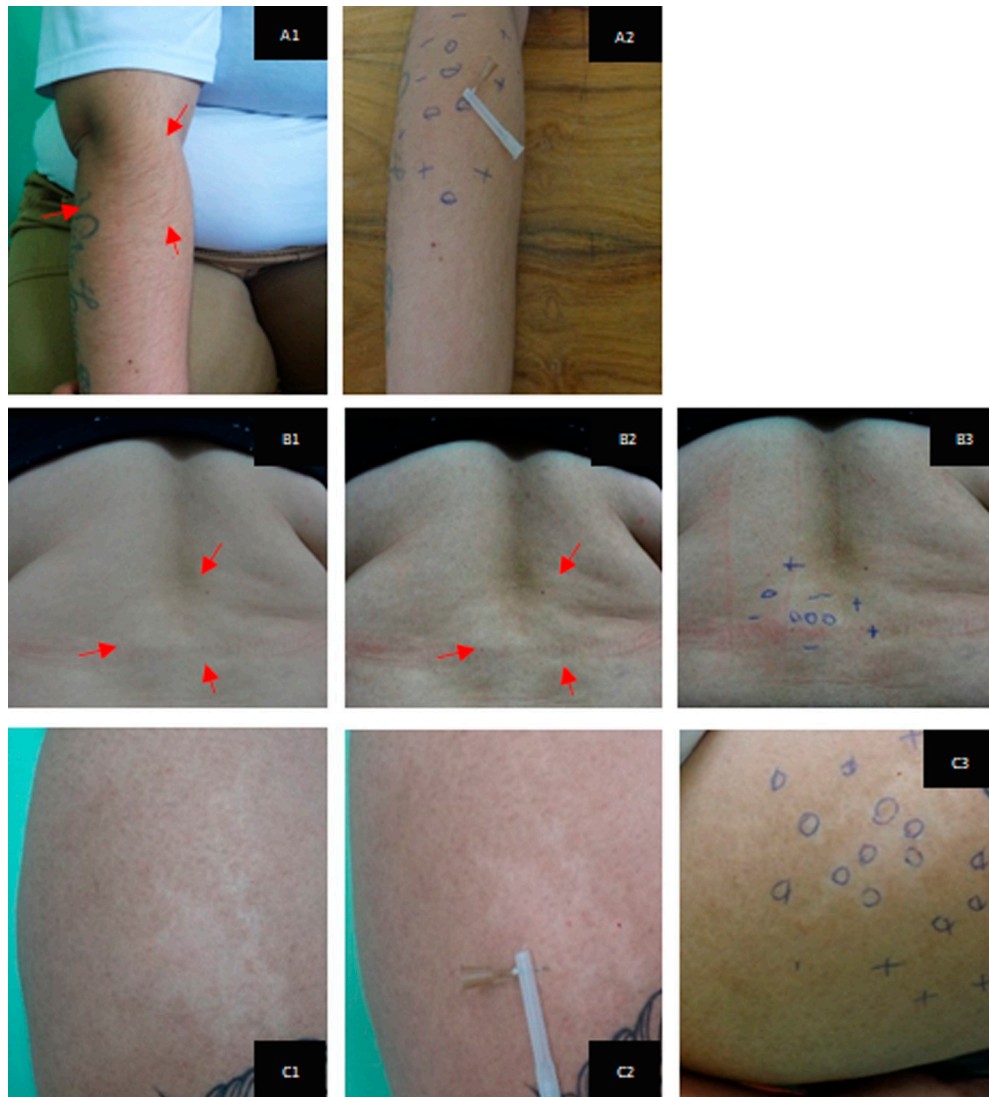

**Fig 1. Clinical-dermatological manifestations due to macular alterations of leprosy, effects of the percentage of contrast applied to the digital images and the respective markings of the results obtained in the sensitivity tests applied.** Hypochromatic macules showing reduction and loss of tactile, thermal and painful sensitivity. Percent image contrast and marking of the responses to the sensitivity test. Reduction of sensitivity [–]; Absence of sensitivity [0]; Normal sensitivity [+] for: A) right forearm; A1- 20% contrast; A2- anesthesia, tactile and painful sensitivity; B) median lumbar region; B1- original image; B2- 40% contrast; B3- tactile sensitivity; C) anterior region of the right thigh; C2 algic anesthesia; C3- tactile sensitivity.

**Table 5. Distribution of anti-PGL-I antibody ELISA index among individuals of Non-Leprosy group (NLG) and Leprosy Group (LG).**

| Groups | TOTAL (n = 404) | | NLG (n = 390) | | LG (n = 14) | | Z | p |
|---|---|---|---|---|---|---|---|---|
| | N | % | N | % | N | % | | |
| Anti-PGL-I < 1 (negative) | 242 | 59.9 | 239 | 61.3 | 3 | 21.4 | -2.98 | 0.003 |
| Anti-PGL-I ≥ 1 (positive) | 162 | 40.1 | 151 | 39.7 | 11 | 78.6 | 2.98 | 0.003 |
| 1.0 |— 1.5 | 65 | 40.1 | 60 | 30.7 | 5 | 45.5 | 2.03 | 0.042 |
| 1.5 |—2.0 | 32 | 19.8 | 30 | 19.9 | 2 | 18.2 | 0.90 | 0.37 |
| ≥ 2.0 | 65 | 40.1 | 61 | 40.4 | 4 | 36.4 | 1.29 | 0.20 |

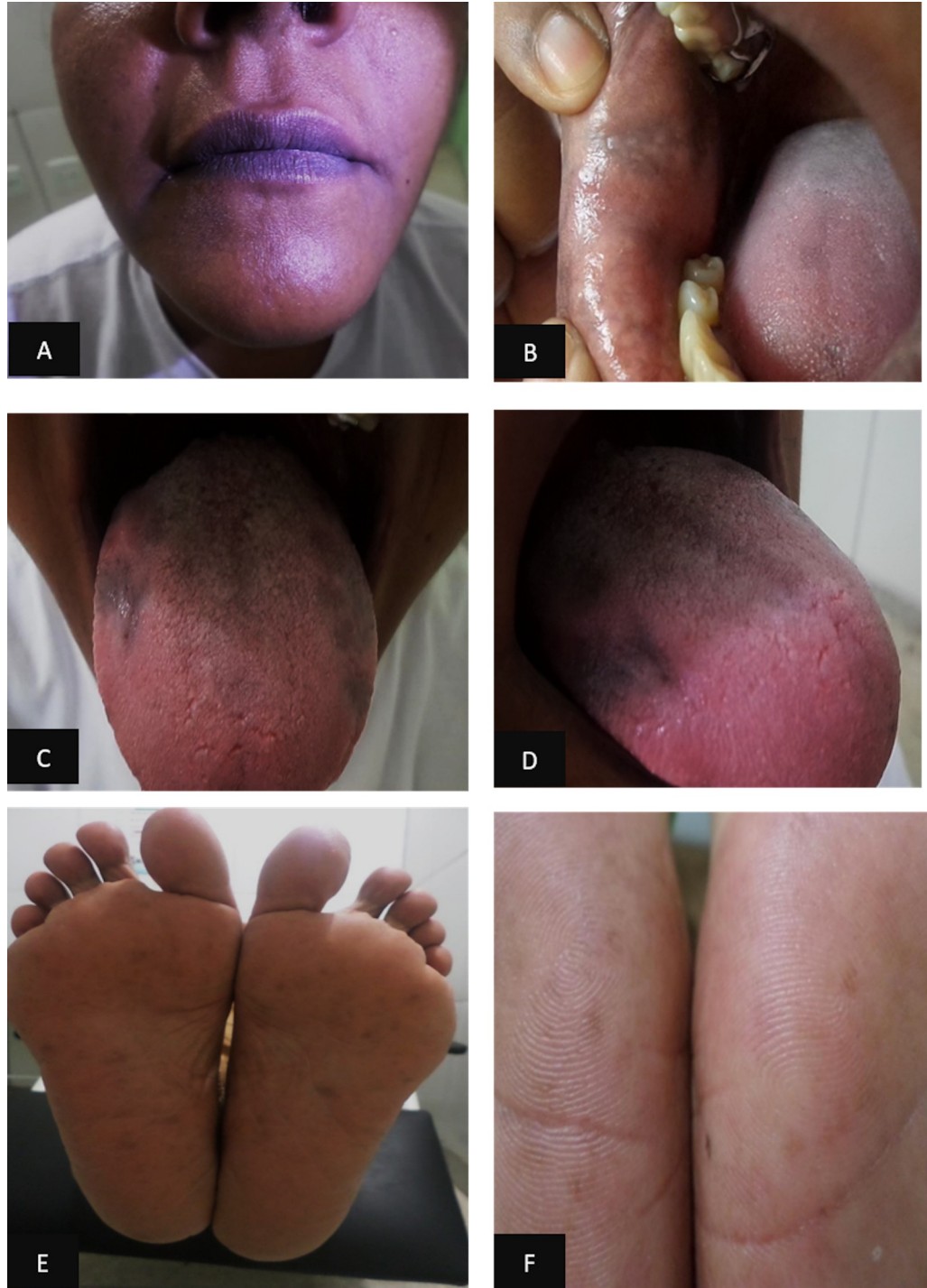

**Fig 2. Clinical manifestations of skin hyperchromia due to the side effect of clofazimine.** (a) Hyperpigmentation of the lip region; (b) internal mucosa of the mouth (cheek); (c) and (d) tongue; (e) plantar, and (f) hypothenar regions.

enanthematous gray lesions mainly on palms and soles and on the lips, internal mucosa of the mouth (cheek) and tongue (Fig 2). In addition, she reported intense headache, epigastralgia, and change in taste, side effects due to clofazimine. The medication was discontinued and

replaced by ofloxacin after 30 days, with a supervised monthly dose of 400 mg and a self-administered daily dose of 400 mg, which was maintained over a period of 12 months. There was a significant improvement of the general symptoms during the first month after the discontinuation of clofazimine with a gradual clearance of skin and mucosal lesions and complete resolution upon discharge.

During the sixth month of treatment, all patients taking MDT for multibacillary leprosy were submitted to a new evaluation of the degree of disability and neural function. One patient taking MDT for paucibacillary leprosy was evaluated in the 3rd and 6th month of treatment, after which she was discharged.

After the 12th month of MDT for multibacillary leprosy, all patients were submitted to a third evaluation of degree of disability and neural function, and a blood sample was collected for anti-PGL-I serology. We observed that 60% of the patients had become negative and 10% of them showed a reduction of the ELISA index.

During the follow-up of 12 patients since their first evaluation, they showed clinical improvement of the disease, and it could be observed by the reports during clinical evaluations and based on esthesiometry at the time of discharge from MDT with significant percentage points in hands and feet regarding the sensitivity to lower weight monofilaments (maximum violet, 2 gram-force) in contrast to the initial assessment with higher weight (maximum black, more than 300 gram-force), as described in Table 6.

In Fig 3, the distribution of the medians of the number of altered esthesiometric points in the hands and feet of the patients confirms the potential of esthesiometry to detect the changes in sensitivity induced by leprosy at diagnosis time (first evaluation), which is greater in the feet than in the hands. Furthermore, the esthesiometry was very objective for measuring therapeutic efficacy based on the recovery of sensitivity as early as in the second evaluation, followed by a more significant effect at the end of MDT in the third evaluation with significance on the hands and feet (p = 0.017 and 0.006 respectively), with a descending tendency line.

## Reevaluation

In view of the large number of anti-PGL-I positive individuals in the NLG (39.7%), a new strategy was adopted in order to reevaluate the female prisoners (n = 66 / 16.3%) who were still in prison 18 months after the first evaluation. The reevaluation showed that 28 women (42.4%) were positive for anti-PGL-I while the other 38 (57.6%) were negative.

**Table 6. Evolution of the number of altered sensitivity points on the hands and feet tested at diagnosis, in the 6th month and after the 12th month of multidrug therapy of the leprosy patients from the Female Penitentiary of Ribeirão Preto.**

| VARIABLE | HANDS | | | FEET | | |
|---|---|---|---|---|---|---|
| | At Diagnosis | 6th MDT | 12th MDT | Diagnosis | 6th MDT | 12th MDT |
| Number of patients in the follow-up LG | 12 | 12 | 11 | 12 | 12 | 11 |
| Total number of changed esthesiometric points | 44 | 22 | 0 | 84 | 68 | 9 |
| Number of patients (%) with altered esthesiometric points | 6 (50.0) | 2 (16.7) | 0 | 10 (83.3) | 8 (66.7) | 5 (45.5) |
| Mean of changed points among those changed | 7.3 | 11 | 0 | 8.4 | 8.5 | 1.8 |
| Mean of altered LG points in follow-up | 3.14 | 1.69 | 0 | 6 | 4.9 | 0.64 |
| Median of altered LG points in follow-up | 0 | 0 | 0 | 2 | 2 | 0 |
| p-value * | | 0.095 | 0.017 | | 0.140 | 0.006 |

LG: leprosy group; MDT: multidrug therapy

*always calculated in relation to the initial values (diagnosis).

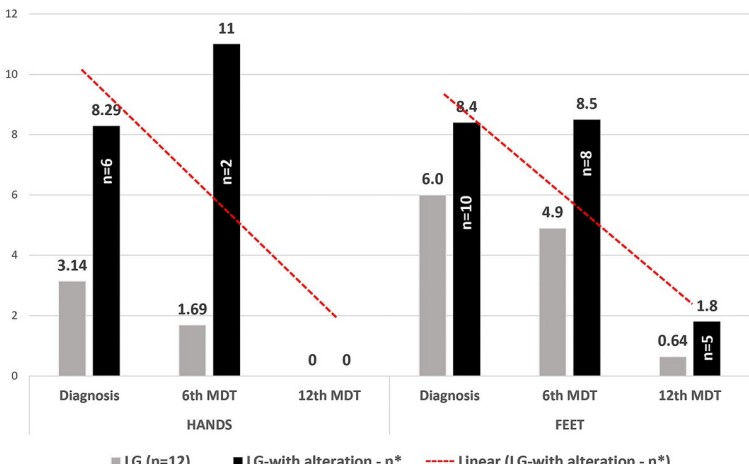

**Fig 3. Averages of the number of altered points in feet and hands in the leprosy group (LG) and among only altered patients (LG–with alteration).** n* = number of patients being followed up in the LG at diagnosis and in the 6th MDT (n = 12) and 12th MDT (n = 11) and patients with altered sensitivity in the hands and feet respectively: n = 7/2/0 in the hands and n = 10/8/5 in the feet.

The LSQ was administered again, and the response rate was 100%, with 25 women (37.9%) marking at least one question. Six questions, Q1, Q2, Q4, Q6, Q7 and Q8, were the most frequently marked in accordance with the results of the previous questionnaire.

Using the same criteria applied on the first occasion, three new cases were diagnosed, all of them multibacillary with a positive LSQ and with an anti-PGL-I positive result.

We also observed that 10 (36%) out of the 28 anti-PGL-I positive subjects had negative results at the second gathering and 8 (30%) showed a reduction of OD positivity values.

Despite these results, after 36 months, we performed a third evaluation of the women remaining in the Prison (n = 14/3, 47%). Three of them (21.4%) had previous anti-PGL-I-positive results and 11 (78.6%) had negative ones. Three new cases were diagnosed, all of them were multibacillary and with marking in the LSQ. One woman was anti-PGL-I positive and the other two were negative.

At the end of the study in this female prison, 20 new cases had been detected and the anti-PGL-I positivity was as high in leprosy (LG) than among those without the disease (NLG) with p-value >0.001 (Table 7). Considering the high serological positivity of the patients to the anti-PGL-I antibody within the female penitentiary, we seek to compare them with the results found among the men of the Penitentiary Progression Center of Jardinópolis described by Bernardes Filho et al [11], as shown in the Fig 4. The medians among total female population and in the NLG were higher than among men. When considering only the female and male leprosy patients, the anti-PGL-I level did not present differences.

There were no new cases of PCR positivity.

**Table 7. Results of measurements of anti-PGL-I antibodies among women tested at the Ribeirão Preto Female Prison (n = 404).**

|  | Total |  | LG |  | LNG |  | *Z* | *P* |
|---|---|---|---|---|---|---|---|---|
| N | 404 | % | 20 | % | 384 | % |  |  |
| Anti-PGL-I < 1 –NEGATIVE | 242 | 60 | 5 | 25 | 233 | 61 |  |  |
| Anti-PGL-I ≥ 1 –POSITIVE | 162 | 40 | 15 | 75 | 151 | 39 | 3.16 | 0.0016 |

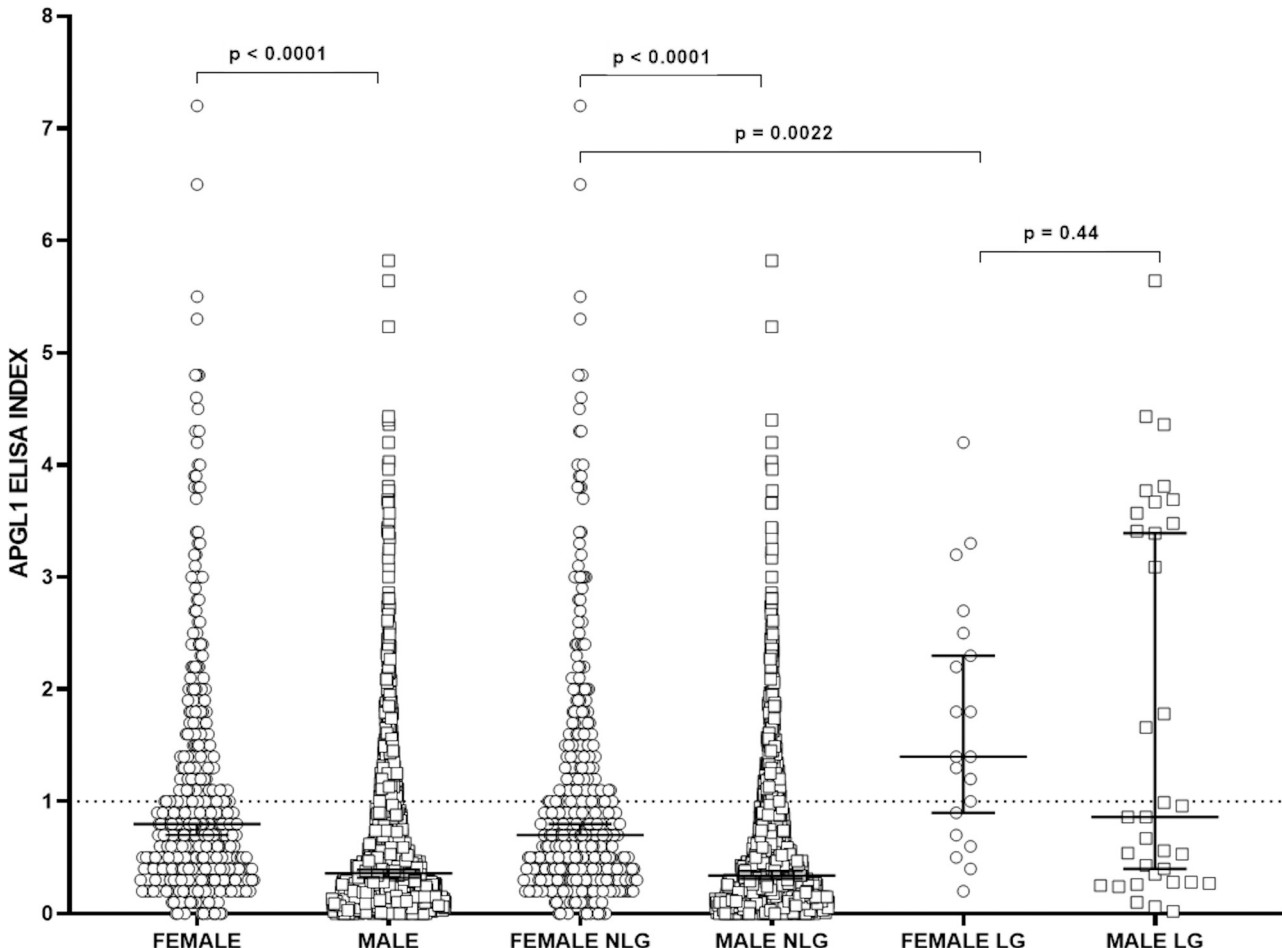

**Fig 4. Comparison of IgM antibody levels by indirect ELISA against PGL-I antigen among all men and women evaluated (Male and Female groups), and among male and female non- leprosy patients (NLP) and all leprosy patients (LP).** Serum levels were compared by the Mann-Whitney test. OD: optical density.

Initially, 14 patients out twenty new leprosy cases (70%) presented altered points on the hands and feet by esthesiometry, six (30%) patients with altered points on the hands, 14 (70%) on the feet, and six (30%) on both.

Esthesiometry of hands and feet revealed that leprosy compromises sensitivity bilaterally, and comparing each point side by side on the hands and feet in the diagnosis, we confirmed the asymmetrical mononeuropathic pattern of the leprosy neuropathy on the feet (p = 0.048); however we did not confirm that on the hands (p = 0.78), as demonstrated in Fig 5. Interestingly, considering as abnormal blue points (0.2-gf) for the hands and violet (2-gf) points for the feet, the total number of altered points per patient was higher on the feet (Σ 124 points / 14 patients / median 1.5) than on the hands (Σ 58 points/7 patients / median 1.18), in which p = 0.003.

All the twenty inmates diagnosed with leprosy were evaluated by ultrasound of peripheral nerves, focusing on the measurements of the largest cross-sectional areas (CSAs) of the analyzed neural points. The results of the medians of the CSAs of the peripheral nerves were: median carpal tunnel—MT (10 mm$^2$), ulnar cubital tunnel UT (6.7 mm$^2$), ulnar pre-cubital tunnel—UPT (6.65 mm$^2$), fibular joint of the fibular head (FCcf) (17.0 mm$^2$), tibial medial

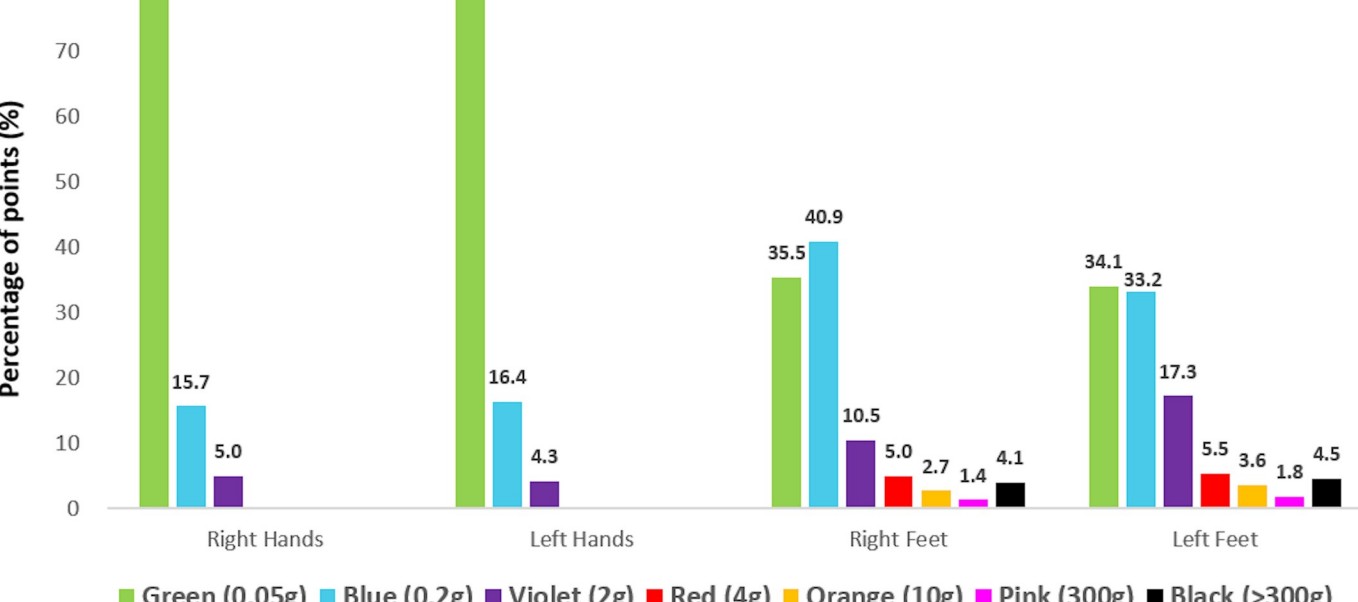

**Fig 5. Distribution of the percentage of numbers of esthesiometric points tested on the right and left hands and feet at the diagnosis of leprosy patients in the Female Prison of Ribeirão Preto, Brazil.**

malleolus (T mm) (11.3 mm$^2$). The upper limits of the normal values of CSA of the peripheral nerves (VR ± 2 SD) were obtained according to Frade et al (2013) [13]: median carpal tunnel (8.9 mm$^2$; 5.9 ± 1.5 mm$^2$); ulnar cubital tunnel (11.1 mm$^2$; 6.7 ± 2.2 mm$^2$); ulnar cubital pre-tunnel (9.5 mm$^2$; 5.9 ± 1.8 mm$^2$) and common fibular in the fibular head (17.0 mm$^2$; 8.2 ± 4.4 mm$^2$). For the tibial nerve, we followed the reference of Qrimli et al, [17] (19.5 mm$^2$; 12.7 ± 3.4 mm$^2$). Additionally, we measured the medial nerve 5mm above carpal tunnel in the forearm–MFo (8.4mm$^2$), and common fibular nerve in the thigh–CFTh (14.8).

Frade et al (2013) [13] reported on the asymmetry between the right and left sides in the same point measuring the difference between the CSAs. The medians of asymmetry were 2.59 mm$^2$ (MT), 1.80 mm$^2$ (MAB), 1.15 mm$^2$ (UT), 1.41 mm$^2$ (UPT), 3 mm$^2$ (CFHf), 2.08 mm$^2$ (CFThigh), and 2.52 mm$^2$ (Tibial). The highest percentage of patients with asymmetry defined as higher than 2mm$^2$ were common fibular nerve on fibular head (55%) and on the median forearm nerve (50%).

Considering the focality among the difference between two points in the same nerve (n = 40 nerves), the medians found in the nerves were: median (1.8 mm$^2$), ulnar (1.0 mm$^2$), and common fibular (1.6 mm$^2$). The focality was higher than 2mm$^2$ in 19 median nerves (47.5%), in common fibular (45%) and only 9 (22.5%) in ulnar nerves.

Electroneuromyographic examination was performed in the 6 new cases diagnosed in the reevaluations. All of them presented an asymmetric and focal multiple mononeuropathy pattern.

## Discussion

The study initially detected 14 new leprosy cases representing 3.5% of the female population followed up at the Female Prison of Ribeirão Preto. Although, the municipality of Ribeirão

Preto has been classified as a high endemic due to the current detection rate of 10.2 per 100 thousand inhabitants [18], this is a considerably high number for so specific population. This value was close to that obtained by Ferreira et al [19], who reported a detection rate of 3.9% within the prison population of the region around Recife, classified as a very high endemic city with a detection rate of 29.2 per 100 thousand inhabitants. Such similar values found in the prisons from the cities with such different classification make these results, at least, paradoxical, mainly when compared to the rates of São Paulo state, which have been under control since 2006 [14]. This fact may be related to the health team involved in the action since all the specialized professionals of the present center (National Reference Center of Sanitary Dermatology–Leprosy, Clinical Hospital of Ribeirão Preto Medical School at, University of São Paulo) are more attuned and trained to recognize a diagnosis of leprosy in its more subtle manifestations besides the additional tools used as LSQ, anti-PGL-I serology. However, it should be pointed out that this detection index was higher than that found for the male population of the same region (2.7%) in a study conducted by the same team at the Center of Penitentiary Progression of Jardinópolis, according to Bernardes Filho et al [11].

As described by Frade et al. [16], most patients had hypochromatic cutaneous macular forms of hypoesthesia and/or anesthetics and were associated with changes in neural trunks, with 95% of the patients being diagnosed with borderline and operationally multibacillary leprosy, demonstrating an advanced stage of the disease among the female inmates. Although the prevalence in Brazil is of medium endemic (1.50 per 10.000 inhabitants), the detection rate is still high and involves a 78.4 percentage of multibacillary cases [20], in accordance with the present findings.

Considering the complex and probably multifactorial transmission of of *M. leprae* and *M. lepromatosis* [21], although no case has been positive on the PCR test in our study, the high proportion of esthesiometry alterations, positive anti-PGL-I serology and multibacillary classification operationally of our patients should be considered as an important *flare gun* of presence and/or transmissibility of the disease to health services among the strategies of active search. Such manifestations and multibacillary classification were found similarly in the male prison population according to data that had already been published by our group [11]. Regarding the lack of positivity in the conventional PCR test for slit skin smears, it may be related to the diagnosis of mild leprosy in our cases that, despite having multiple sites of nervous involvement, these cases seem to have a low bacillary load.

The LSQ is a tool employed in the active search promoted by our Reference Center, with more than 7.000 questionnaires administered, including the male prison population [11]. As stated above, the general index of new cases in the male prison was 2.7%; however, among the patients who had some markings in the questionnaire, the index was 9.6% demonstrating the importance of the LSQ for the screening of new leprosy cases [11]. In the present study, 100% of the patients had some markings, and when more than one question was marked, there was a significant 13-fold increase in the occurrence of the illness, confirming the effectiveness of the LSQ. When we applied the LSQ, we observed similarities between the two groups; 4 out of the 6 questions the most frequently marked denoted neural symptoms, as also observed in the male prison population (3 out of the 4 questions the most frequently marked denoted neural symptoms) [11], supporting the importance of a clinical diagnosis based on the observation of the neural symptoms of the disease.

Applying the chi-square test to assess the relationship between each question individually to find the possibility of the patients belonging or not to the leprosy group, the highest results with Yates correction were 34.5 for pain in the nerves (Q7), 7.5 for anesthetized areas in the skin (Q3), 5.57 for numbness on the hands or feet (Q1), and 5.77 for stinging sensation (Q5), reaching the latest significant place of the chi-square value, the Q4 about spots on the skin

(4.94), appearing in 5th place, showing once again, how important the neurological symptoms are rather than the cutaneous signs for the diagnosis of leprosy. Corroborating with the importance of the neurological symptoms for the diagnosis of leprosy, we got the highest odd ration when the symptom stinging sensation (Q5) was associated to other neurological symptoms, as numbness (Q1), tingling (Q2), and pain in the nerves (Q7).

We also observed that, although the disease was classified as multibacillary due to the expertise of the team and to the diagnostic arsenal used, these manifestations are quite discrete and would easily go unperceived by basic care teams caring for the prison population.

The neural status is the most important outcome for an early diagnosis of the disease considering that the risk of neural injury will involve definitive sequelae [22]. In accordance with this statement, there are the results obtained by Santos et al [23], who emphasized the importance of neural thickening as a diagnostic suspicion of leprosy in regions endemic for the disease.

All new patients who were kept in prison were monitored and concluded the multidrug therapy, being discharged from treatment within the predicted time and showing significant clinical improvement. The esthesiometer was used both to aid the diagnosis of sensitivity changes with significant relation with anti-PGL-I positivity, and to identify the improvement of sensitivity. According to a study by Santos et al. [24], for the characterization of the degree and symmetry of neural involvement in leprosy patients, the use of monofilaments is a strategy for the prevention of severe and often irreversible deformities in leprosy patients.

The use of an esthesiometer for the early diagnosis and the monitoring of peripheral injury is an important strategy that permits the detection of improvement of the changes in sensitivity since the evaluation of sensitivity is fundamental for the diagnosis of neuropathy [16]. The esthesiometer was essential for the assessment of the degree of disability determined on three occasions: at diagnosis, with the degree of physical disability (DPD I) inferred in 70% of the patients during the sixth month of MDT, and upon discharge, when DPD I was equal to zero. Although the DPD I demonstrated that disease detection was late, the progression to DPD equal to zero and the reports of clinical improvement, especially regarding neurological and sensitivity complaints, emphasized the results obtained at the end of treatment and also confirmed the bacterial etiology of the neuropathy since only a specific antimicrobial treatment was used.

The evaluation of the percentage of number of esthesiometric points in the feet over the three assessments (at diagnosis, after six months of treatment and at discharge) revealed that the sum of the green and blue points initially was 72% in the left foot and 63% in the right one, with points varying up to pink (300-gf) and black (>300-gf). However, at the final evaluation, the green and blue points represented 97% of the points in the right foot and 95% in the left foot, with only violet points (2-gf) continuing to be abnormal showing a significant improvement. And in the hands, all points became green and returned to normality. At discharge, there was a minimum pattern of alteration that corresponded to a cure. This was not simply a microbiological cure, but also a functional cure, in accordance with Barreto et al [4] who argued that MDT breaks the chain of transmission of the disease and avoids progression, preventing disabilities and deformities, reestablishing sensitivity and providing cure.

We observed and demonstrated that changes were greater in the lower limbs than in the upper limbs, even though most published studies report that the ulnar nerve is the most affected one. In a study on leprosy neuropathy, Van Brakel et al. [25] detected a higher prevalence of ulnar involvement, as also reported by Lugão [26], who stated that the ulnar nerve was the one most frequently and intensely affect in multibacillary patients.

We want to emphasize that Nascimento et al [27] identify the ulnar nerve as the most evident nerve in leprosy patients. In this respect, we believe that the greatest focus on the ulnar

nerve in these studies may be due to the fact this nerve is located between medial epicondyle of humerus and olecranon, in an superficial area which is more accessible to traumas in terms of daily movements and activities, in contrast to the fibular nerve located behind of fibular head and tibial nerve located posterior to medial malleolus. Another important factor is that the posterior tibial nerve is not easily nor routinely evaluated in electroneuromyography and there is a few number of papers about its impairment in leprosy [28, 29]. Therefore, our study cannot be compared to others since the proposed esthesiometry only measures plantar sensitivity fully determined by the fibers of the tibial nerve. Thus, using esthesiometry and Semmes-Weinstein monofilaments, we were able to determine that the most affected nerve in leprosy is the tibial because of the greater number of points involved. However, we support the importance of the evaluation of all nerves in the routine assessment of patients with suspected leprosy.

In this study, we used ultrasonography, a technique that allows a good quantitative and qualitative representation of peripheral, superficial and deep nerves by measuring transverse sectional areas that are important for the diagnosis of neuropathy and the detection of peripheral nerve thickening in leprosy, as reported by Frade et al [13]. Ultrasound evaluation confirmed the multibacillary classification of the patients since 95% of them showed nerve thickening. According to Lugão et al. [25], a clinical exam in combination with ultrasound can identify a greater dimension of the nerve and its neurological involvement.

Interestingly, the thickening of the median and common fibular nerves was identified among leprosy patients. It is reiterated that the median nerve is located within an osteofibrous tunnel, which may be correlated with other more frequent causes of neuropathy, and it has not shown significant changes in the clinical evaluation. On the other hand, among leprosy patients in this study, 50% presented asymmetry on the forearm neural point at the diagnosis. Therefore, the common fibular stands out as a possible marker of early diagnosis of leprosy neuropathy, since the ulnar nerve, considered the main nerve to be altered in leprosy has not yet thickened.

During MDT, 8 patients had side effects such as abdominal pain, headache and nausea, with darkening of the skin and mucosae occurring in one case. Some side effects can be observed during MDT [27]; Rifampin and dapsone are usually responsible for the intestinal and digestive complaints and for the headache, and the main side effect of clofazimine is skin dryness and reddish or brownish pigmentation, in addition to adverse gastrointestinal effects [30, 31].

The treatment of the disease also led to autonomic dysfunctions, with xeroderma in 29.4% of the patients and ocular dryness in 35.3%. These are signs and symptoms requiring care that were also observed by Koshy et al. [32], with high percentage of leprosy patients at risk for dry eye and for impaired skin integrity.

In the present prison population, the BCG scar was absent in only 10.7% of the inmates, but it was present in 15% of the patients diagnosed for leprosy, without indicating a greater risk for the disease. These findings disagree with those reported by Goulart et al. [33], who reported a greater risk of about 3.7 times regarding the occurrence of leprosy among individuals not vaccinated with BCG.

In the literature, the 31-moth average detention time is shorter than the time for the incubation of the disease, which is long, with an interval of 2 to 7 years reported by the Ministry of Health [12] and of 5 to 10 years regarding other aspects, such as relationship with the host and degree of environmental endemic [1]. According to Silva et al. [34], the incubation of *M. leprae* is exceptionally long; it takes the disease more than ten years to manifest in some cases. Thus, we believe that in the present study, transmissibility may have occurred before detention, as

also reported for the male prison population [11]. Such a fact was also observed by Ferreira et al [19] who reported a 21-month detention time for women detected with leprosy.

In the serological analysis for anti-PGL-I, their ELISA indexes were significantly higher among women in confinement (median 0.800), more than double when compared to that found in the male penitentiary (median 0.360) [11]. The 40% positivity of the anti-PGL-I index in the prison population is high and, even with a median of 0.70 and meaning half the median of patients of 1.4, the data epidemiologically suggest the presence of the bacillus in this population, in accordance with literature findings demonstrating that untreated multibacillary patients from endemic areas show higher positivity for IgM anti-PGL-I [35, 36].

Another important aspect regarding the anti-PGL-I high positivity is the possibility of sub-clinical infection with *M. leprae* [1, 37]. The anti-PGL-I positive women studied herein were reevaluated clinically after 18 months (n = 66) and 36 months (n = 14), with six more cases being detected among them, supporting the relationship described by the above authors, as well as the importance of anti-PGL-I serology in the screening and active search of new cases of leprosy. We also observed that the serology performed at discharge showed that 60% of the patients had become negative and 10% had a reduced ELISA index, demonstrating the relationship between the use of MDT and the break of transmissibility, an essential action to interrupt the chain of disease transmission, reflecting directly on the anti-PGL-I serological levels over time.

## Conclusion

The active search action found a significant leprosy new case detection rate inside the Female Prison, and our data confirmed a hidden endemic and also demonstrated that the transmission probably occurred before detention, considering the time of imprisonment was shorter than that of incubation of the disease.

Like in the male prison population, LSQ proved to be an auxiliary screening tool in the active search among female since all new cases showed some marking, mainly the neurological symptoms, as Q7+ (pain in the nerves) and Q5+ (stinging sensation) which were individually analyzed. Therefore, when Q5+ was marked along with Q1+ (numbness), Q7+, and Q2+ (tingling), we got the highest odds ration as 10.1, 7.5 and 8.1 respectively, making it, instead of skin signals, a warn sign for the diagnosis of leprosy.

The high serum positivity of anti-PGL-I epidemiologically demonstrates the prison population contact with the bacillus. Among leprosy patients, their positivity was related with hands and feet tactile sensitivity (esthesiometer), important for the diagnosis and monitoring of the peripheral lesions of leprosy, which is an important strategy that allows the identification of injured areas and the improvement of sensitivity changes.

Finally, the study permitted the evaluation of the efficacy of an active search of cases inside the prison with our instruments for clinical-epidemiological evaluation, such as the LSQ, the use of a specific dermato-neurological exam, and screening and serological follow-up for anti-PGL-I, all of them important for the definition of the risk of getting ill and for better strategies for the promotion of care and treatment of the prisoners, making it possible to break the chain of transmission within the prison and to reestablish the health of those who leave the penal system.

## Author Contributions

**Conceptualization:** Claudia Maria Lincoln Silva, Fred Bernardes Filho, Marco Andrey Cipriani Frade.

**Data curation:** Claudia Maria Lincoln Silva, Fred Bernardes Filho, Natália Aparecida de Paula, Patricia Toscano Barreto Nogueira Onofre, Wilson Marques-Junior, Marco Andrey Cipriani Frade.

**Formal analysis:** Marco Andrey Cipriani Frade.

**Funding acquisition:** Marco Andrey Cipriani Frade.

**Investigation:** Claudia Maria Lincoln Silva, Fred Bernardes Filho, Glauber Voltan, Jaci Maria Santana, Marcel Nani Leite, Filipe Rocha Lima, Luisiane de Avila Santana, Natália Aparecida de Paula, Patricia Toscano Barreto Nogueira Onofre, Wilson Marques-Junior, Vanessa Aparecida Tomaz, Carmem Sílvia Vilela Pinese, Marco Andrey Cipriani Frade.

**Methodology:** Marco Andrey Cipriani Frade.

**Project administration:** Marco Andrey Cipriani Frade.

**Resources:** Marco Andrey Cipriani Frade.

**Supervision:** Marco Andrey Cipriani Frade.

**Visualization:** Marco Andrey Cipriani Frade.

**Writing – original draft:** Fred Bernardes Filho, Marco Andrey Cipriani Frade.

**Writing – review & editing:** Claudia Maria Lincoln Silva, Fred Bernardes Filho, Natália Aparecida de Paula, Marco Andrey Cipriani Frade.

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
