## [Decision Letter · Decision Letter 0]

7 Jul 2021

Dear Dr Frade,

Thank you very much for submitting your manuscript "Innovative tracking, active search and follow-up strategies for new leprosy cases in the female prison population" for consideration at PLOS Neglected Tropical Diseases. As with all papers reviewed by the journal, your manuscript was reviewed by members of the editorial board and by several independent reviewers. The reviewers appreciated the attention to an important topic. Based on the reviews, we are likely to accept this manuscript for publication, providing that you modify the manuscript according to the review recommendations. 

Sincerely,

Carlos Franco-Paredes

Associate Editor

Richard Phillips

Deputy Editor

Reviewer's Responses to Questions

**Key Review Criteria Required for Acceptance?**

**Methods**

-Are the objectives of the study clearly articulated with a clear testable hypothesis stated?

-Is the study design appropriate to address the stated objectives?

-Is the population clearly described and appropriate for the hypothesis being tested?

-Is the sample size sufficient to ensure adequate power to address the hypothesis being tested?

-Were correct statistical analysis used to support conclusions?

-Are there concerns about ethical or regulatory requirements being met?

Reviewer #1: The objectives of the study are clear. But what they want to proof you can doubt. Population is well descibed and appropiate to test the hypothesis. But they try to do too much

Reviewer #2: -Are the objectives of the study clearly articulated with a clear testable hypothesis stated? - Yes

-Is the study design appropriate to address the stated objectives? - Yes

-Is the population clearly described and appropriate for the hypothesis being tested? - Yes

-Is the sample size sufficient to ensure adequate power to address the hypothesis being tested? - Yes

-Were correct statistical analysis used to support conclusions? - Yes

-Are there concerns about ethical or regulatory requirements being met? - No

**Results**

-Does the analysis presented match the analysis plan?

-Are the results clearly and completely presented?

-Are the figures (Tables, Images) of sufficient quality for clarity?

Reviewer #1: yes.

Reviewer #2: The analysis, results and tables/figures are well presented.

**Conclusions**

-Are the conclusions supported by the data presented?

-Are the limitations of analysis clearly described?

-Do the authors discuss how these data can be helpful to advance our understanding of the topic under study?

-Is public health relevance addressed?

Reviewer #1: The do that well

Reviewer #2: Authors should check the Summary and General Comments.

**Editorial and Data Presentation Modifications?**

Reviewer #1: Rewrite.

Reviewer #2: There is a need for spelling corrections.

**Summary and General Comments**

Reviewer #1: Dear Authors,

Having read your paper on using your questionnaire in town and in a male prison, I now see it applied in a female prison. What is new? Are there differences is it a repetition or something new? 

It is a beautiful report of a careful done survey. But more a thesis then a paper. As clinician I enjoy reading it. But hardly anything new. Only it made me more critical on the LSQ. I wonder whether except for testing this test, a prison is the right place were it is needed.

What is new? 

Just a few comments going through the paper:

I would have the English checked.

The paper certainly is of local interest and findings such a great number of patients with leprosy shows that the study was worth doing and makes you wonder what of the real number of leprosy cases is in underprivileged areas.

You mention that 3 patients were diagnosed 18 and 36 months later. Why were they missed before? What extra signs did develop? This is something you may learn from. 

Even in healthy persons the lower limb has lower values then the upper limbs. What was the criteria to call them affected?

You evaluated all 404 inmates. Was there an incentive? 

59 you did not revaluate the patients but the inmates. These 6 were found after how long time? Are these different from the 3 after 18 and 36 months? 

Were the 6 not suspected after LSQ. How many would be discovered when only the positive LSQ persons were evaluated. Why is it more efficient with the LSQ when you evaluate all inmates?

The port of exit of the bacilli is most likely upper respiratory tract, but port of entre may be different.

76 over crowding and precarious condition ( what does that imply?) Maybe you mean what is written in the next sentence. 

All got the questionnaire, and all were examined. What was the contribution of the LSQ? Except assessing the questionnaire again. 

The nerve ultrasonography was a separate research or did it contribute to the diagnosis. Was there any patient diagnosed that was missed otherwise? 

Same question for anti-PGL1. 

And in a way also for the DNA extraction from the smear. Did you compare between microscopy and Staining for AFB? 

You say the sensory testing was done at start and 6 month and at the end. I may hope also when needed.

In table 1 check the lines in the last 2 columns.

How large do you expect is the chance of having leprosy with a positive LSQ compared with a positive anti PGL-1? You first paper by using it in town give me a good feeling. With the male prisoners I still thought it a contribution. But now I start to doubt. Statistics leave out individuals. Because like positive or negative anti PGL-1 you only have an indication. And then anti-PGL-1 can be used for epidemiology and LSQ not.

218 The longer the peripheral nerves the more chance you have to find abnormalities: leprosy is not different from diabetes. PGL-1 by itself has been shown to be able to damage the myeline sheath.

On 243 you mention that the patients were measured monthly, as it should with esthesiometry evaluations. Before you mentioned 6 and 12 months.

The discussion is very long this is mainly because you studied so many parameters.

The “thesis” could be separated in smaller papers and published in appropriate journals

Reviewer #2: The results of this work are very interesting and can contribute to the early diagnosis of leprosy. Some aspects caught the attention of this reviewer, highlighted below:

• In the abstract:

“The NCDR in this population showed hidden leprosy like among males in prison, as

well as the efficacy of a search action on the part of a specialized team with the aid of

the LSQ as an auxiliary tracking tool”.

The author compares the results obtained from this study with a similar study carried out by the research team in the imprisoned male population. However, this consideration in the abstract is inappropriate. In the discussion of the results, this aspect would be relevant.

• In the discussion:

Lines 371-374: “This fact may be related to the quality of the health team involved in the action since all the specialized professionals of the present center (National Reference Center of Sanitary Dermatology – Leprosy, Clinical Hospital of Ribeirão Preto Medical School at, University of São Paulo) are more attuned and trained to recognize a diagnosis of leprosy in its more subtle manifestations”.

 Despite the different epidemiological scenarios between the two regions, (São Paulo and Recife), it must be considered that the methodology used for the two studies are different and that no value judgment is appropriate in relation to the clinical expertise of the authors involved in the study cited as a reference.

 Furthermore, considering leprosy a "hidden endemic" in a country with continental dimensions such as Brazil, it is possible to analyse these results from another aspect, such as the lack of expertise of professionals in the region where this same study was carried out.

 The well-trained team of this work, under revision using a different and more accurate methodology, arrived at results close to this same group mentioned, which used the most common and routinely available tools in the field [1].

Lines 383-386: “The high proportion of multibacillary patients observed in our study should be considered as an important flare gun of transmissibility of the disease to health services in the strategies of active search and consequently of breaking the chain of transmission, since the multibacillary form, presents in 95 % of the diagnosed cases, is recognized by its great power of transmission”. (There are spelling that needs correction). 

 It is possible that this conclusion was reinforced if there were parameters for bacilloscopic evaluation, which is a gap in this study, also in relation to PCR’s positivity. (The authors’ statement in line 323 is that “There were no new cases of PCR positivity”) Nevertheless, it is true that early diagnosis occurred in the studied groups, using esthesiometry and anti-PGL-I serology.

 However, the transmission of leprosy among cases classified as multibacillary so far is described as being possible due to the elimination of bacilli through the known routes of this group of patients to the outside world.

Although leprosy, according to the most recent WHO guidelines, can be classified as multibacillary with the detection of only one compromised nerve, the assertion that these patients listed is this study (at this stage of the disease) are responsible for maintaining the epidemiological chain of transmission can be hasty, for the time. The authors seem to understand this aspect, in lines 409-411. The leprosy transmission is complex is multifactorial.

 According Ploemacher and cols (2020): “This entails integrating human, animal, and environmental health aspects to further elucidate the transmission mechanisms and patterns of M. leprae and M. lepromatosis. In addition, geographically tailored methods–combining epidemiological, laboratory, and anthropologic data–are needed to better understand the ecological differences between leprosy pockets” [2]. 

 Once they realized, the authors need to clarify or discuss the lack of positivity in the PCR test.

Lines 449-451: “We believe that the greatest focus on the ulnar nerve in these studies may be due to the fact this nerve is located in an area that is more accessible to traumas in terms of daily movements and activities, in contrast to the fibular and tibial nerves”.

 This statement deserves attention, considering that the plantar regions whose innervation is the responsibility of the posterior tibial nerve are as susceptible (or perhaps more susceptible) to trauma than those innervated by the ulnar nerve. There are several studies that offer attention to the posterior tibial nerve in leprosy [3,4,5,6].

 This work is relevant and looks at a neglected disease in a neglected population. It demonstrates that simple tools such as an esthesiometer and a well-applied questionnaire can significantly contribute to the diagnosis of leprosy. To corroborate their results, however, the authors use sophisticated tools that are unavailable to the vast majority, needing to discuss this important aspect.

References:

1. Bernardes F Filho, Paula NA, Leite MN, et al. Evidence of hidden leprosy in a supposedly low endemic area of Brazil. Mem Inst Oswaldo Cruz. 2017;112(12):822-828. doi:10.1590/0074-02760170173.

2. Ploemacher T, Faber WR, Menke H, Rutten V, Pieters T. Reservoirs and transmission routes of leprosy; A systematic review. PLoS Negl Trop Dis. 2020;14(4):e0008276. Published 2020 Apr 27. doi:10.1371/journal.pntd.0008276.

3. Gupta BK, Kochar DK. Study of nerve conduction velocity, somatosensory-evoked potential and late responses (H-reflex and F-wave) of posterior tibial nerve in leprosy. Int J Lepr Other Mycobact Dis. 1994;62(4):586-593.

4. Richard B, Khatri B, Knolle E, Lucas S, Turkof E. Leprosy affects the tibial nerves diffusely from the middle of the thigh to the sole of the foot, including skip lesions. Plast Reconstr Surg. 2001;107(7):1717-1724. doi:10.1097/00006534-200106000-00012.

5. Jain S, Visser LH, Praveen TL, et al. High-resolution sonography: a new technique to detect nerve damage in leprosy. PLoS Negl Trop Dis. 2009;3(8):e498. Published 2009 Aug 11. doi:10.1371/journal.pntd.0000498.

6. Wagenaar I, Post E, Brandsma W, et al. Early detection of neuropathy in leprosy: a comparison of five tests for field settings. Infect Dis Poverty. 2017;6(1):115. Published 2017 Sep 1. doi:10.1186/s40249-017-0330-2.

PLOS authors have the option to publish the peer review history of their article (what does this mean?). If published, this will include your full peer review and any attached files.

Reviewer #1: No

Reviewer #2: No

Figure Files:

Data Requirements:

Reproducibility:

References

---

## [Editor Report · Decision Letter 1]

7 Aug 2021

Dear Dr Frade,

We are pleased to inform you that your manuscript 'Innovative tracking, active search and follow-up strategies for new leprosy cases in the female prison population' has been provisionally accepted for publication in PLOS Neglected Tropical Diseases.

Best regards,

Carlos Franco-Paredes

Associate Editor

Richard Phillips

Deputy Editor

---

## [Editor Report · Acceptance letter]

17 Aug 2021

Dear Dr Frade,

We are delighted to inform you that your manuscript, "Innovative tracking, active search and follow-up strategies for new leprosy cases in the female prison population," has been formally accepted for publication in PLOS Neglected Tropical Diseases.

Best regards,

Shaden Kamhawi

co-Editor-in-Chief

Paul Brindley

co-Editor-in-Chief
